# Determining Chess Game State from an Image

**DOI:** 10.3390/jimaging7060094

**Published:** 2021-06-02

**Authors:** Georg Wölflein, Ognjen Arandjelović

**Affiliations:** School of Computer Science, University of St Andrews, North Haugh, St Andrews KY16 9SX, Scotland, UK; oa7@st-andrews.ac.uk

**Keywords:** computer vision, chess, convolutional neural networks

## Abstract

Identifying the configuration of chess pieces from an image of a chessboard is a problem in computer vision that has not yet been solved accurately. However, it is important for helping amateur chess players improve their games by facilitating automatic computer analysis without the overhead of manually entering the pieces. Current approaches are limited by the lack of large datasets and are not designed to adapt to unseen chess sets. This paper puts forth a new dataset synthesised from a 3D model that is an order of magnitude larger than existing ones. Trained on this dataset, a novel end-to-end chess recognition system is presented that combines traditional computer vision techniques with deep learning. It localises the chessboard using a RANSAC-based algorithm that computes a projective transformation of the board onto a regular grid. Using two convolutional neural networks, it then predicts an occupancy mask for the squares in the warped image and finally classifies the pieces. The described system achieves an error rate of 0.23% per square on the test set, 28 times better than the current state of the art. Further, a few-shot transfer learning approach is developed that is able to adapt the inference system to a previously unseen chess set using just two photos of the starting position, obtaining a per-square accuracy of 99.83% on images of that new chess set. The code, dataset, and trained models are made available online.

## 1. Introduction

The problem of recovering the configuration of chess pieces from an image of a physical chessboard is often referred to as chess recognition. Applications span chess robots, move recording software, and digitising chess positions from images. A particularly compelling application arises in amateur chess, where a casual over-the-board game may reach an interesting position that the players may afterwards want to analyse on a computer. They acquire a digital photograph before proceeding with the game, but once the game concludes, they must enter the position piece by piece on the computer—a process that is both cumbersome and error-prone. A system that is able to map a photo of a chess position to a structured format compatible with chess engines, such as the widely-used Forsyth–Edwards Notation (FEN), could automate this laborious task.

To this end, we put forth a new synthesised dataset [1] comprising of rendered chessboard images with different chess positions, camera angles, and lighting setups. Furthermore, we present a chess recognition system consisting of three main steps: (i) board localisation, (ii) occupancy classification, and (iii) piece classification. For the latter two steps, we employ two convolutional neural networks (CNNs), but make use of traditional computer vision techniques for board localisation.

However, chess sets vary in appearance. By exploiting the geometric nature of the chessboard, the board localisation algorithm is robust enough to reliably recognise the corner points of different chess sets without modification. Using this algorithm in conjunction with careful data augmentation, we can extract sufficient samples from just two unlabelled photos of a previously unseen chess set (in the starting position) for fine-tuning the CNNs to adapt the system for inference on that new chess set. The code and trained models are available at https://github.com/georgw777/chesscog, accessed on 30 May 2021.

## 2. Previous Work

Initial research into chess recognition emerged from the development of chess robots using a camera to detect the human opponent’s moves. Such robots typically implement a three-way classification scheme that determines each square’s occupancy and (if occupied) the piece’s colour [2,3,4,5,6,7]. Moreover, several techniques for recording chess moves from video footage employ the same strategy [8,9,10]. However, any such three-way classification approach requires knowledge of the previous board state to deduce the current chess position (based on the last move inferred from its predictions of each square’s occupancy and piece colour). While this information is readily available to a chess robot or move recording software, it is not for a chess recognition system that receives just a single still image as input. Furthermore, these approaches are unable to recover once a single move was predicted incorrectly and fail to identify promoted pieces (piece promotion occurs when a pawn reaches the last rank, in which case the player must choose to promote to a queen, rook, bishop or knight; vision systems that can only detect the piece’s colour are unable to detect what it was promoted to).

A number of techniques have been developed to address the issue of chess recognition from a single image by classifying each piece type (pawn, knight, bishop, rook, queen, and king) and colour, mainly in the last five years. Since chess pieces are nearly indistinguishable from a bird’s-eye view, the input image is usually taken at an acute angle to the board. While Ding [11] relies on scale-invariant feature transform (SIFT) and histogram of oriented gradients (HOG) feature descriptors for piece classification, Danner et al. [12] as well as Xie et al. [13] claim that these are inadequate due to the similarity in texture between chess pieces, and instead apply template matching to the pieces’ outlines. However, Danner et al. modify the board colours to red and green instead of black and white to simplify the problem (similar modifications have also been proposed as part of other systems [3,9]), but any such modification imposes unreasonable constraints on normal chess games.

Several other techniques have been developed that employ CNNs at various stages in the recognition pipeline. Xie et al. compare their template matching approach to the use of CNNs as part of the same work, finding only minor increases in accuracy (though they trained on only 40 images per class). Czyzewski et al. [14] achieve an accuracy of 95% on chessboard localisation from non-vertical camera angles by designing an iterative algorithm that generates heatmaps representing the likelihood of each pixel being part of the chessboard. They then employ a CNN to refine the corner points that were found using the heatmap, outperforming the results obtained by Gonçalves et al. [7]. Furthermore, they compare a CNN-based piece classification algorithm to the SVM-based solution proposed by Ding [11] and find no notable amelioration, but manage to obtain improvements by reasoning about likely and legal chess positions. Recently, Mehta et al. [15] implemented an augmented reality app using the popular AlexNet CNN architecture [16], achieving promising results. Their CNN was trained to distinguish between 13 classes (six types of pieces in either colour, and the empty square) using a dataset of 200 samples per class. Despite using an overhead camera perspective, they achieve a per-square accuracy of 93.45% on the end-to-end pipeline (corresponding to a per-square error rate of 6.55%), which – to the best of our knowledge – constitutes the current state of the art.

A prerequisite to any chess recognition system is the ability to detect the location of the chessboard and each of the 64 squares. Once the four corner points have been established, finding the squares is trivial for pictures captured in bird’s-eye view, and only a matter of a simple perspective transformation in the case of other camera positions. Some of the aforementioned systems circumvent this problem entirely by prompting the user to interactively select the four corner points [5,7,8,12], but ideally a chess recognition system should be able to parse the position on the board without human intervention. Most approaches for automatic chess grid detection utilise either the Harris corner detector [3,10] or a form of line detector based on the Hough transform [4,6,12,17,18,19,20], although other techniques such as template matching [21] and flood fill [9] have been explored. In general, corner-based algorithms are unable to accurately detect grid corners when they are occluded by pieces, thus line-based detection algorithms appear to be the favoured solution. Such algorithms often take advantage of the geometric nature of the chessboard which allows to compute a perspective transformation of the grid lines that best matches the detected lines [10,13,17].

Adequate datasets for chess recognition—especially at the scale required for deep learning—are not available as of now, an issue that has been recognised by many [11,14,15]. To this end, synthesising training data from 3D models seems to be a promising avenue to efficiently generate sizable datasets while eliminating the need for manual annotation. Wei et al. [22] synthesise point cloud data for their volumetric CNN directly from 3D chess models and Hou [23] uses renderings of 3D models as input. Yet, the approach of Wei et al. works only if the chessboard was captured with a depth camera and Hou presents a chessboard recognition system using a basic neural network that is not convolutional, achieving an accuracy of only 72%.

## 3. Dataset

Studies in human cognition show that highly skilled chess players generally exhibit a more developed pattern recognition ability for chess positions than novices, but this ability is specific to positions that conform to the rules of chess and are likely to occur in actual games [24]. To ensure that the chess positions in our synthesised dataset are both legal and sensible, we randomly sample 2% of all positions (i.e., configurations of chess pieces) from a public dataset of 2851 games played by current World Chess Champion Magnus Carlsen. After eliminating duplicates, a total of 4888 chess positions are obtained in this manner, saved in FEN format, and split into the training (90%), validation (3%), and test (7%) sets.

In order to obtain realistic images of these chess positions, we employ a 3D model of a chess set on a wooden table. Chess pieces are placed on the board’s squares according to each FEN description, but are randomly rotated around their vertical axis and positioned off-centre according to a normal distribution to emulate real conditions. Different camera angles (between 45° and 60° to the board) and lighting setups (either randomly oriented spotlights or a simulated camera flash) are chosen in a random process to further maximise diversity in the dataset, as depicted in Figure 1. The 4888 positions are rendered in an automated process. With each image, we export as labels the FEN description of the position as well as the pixel coordinates of the four corner points.

The dataset contains 104,893 samples of squares occupied by a piece and 207,939 empty squares. In the occupied squares, the most frequent class are black pawns with 27,076 occurrences and the least frequent are black queens with 3133 samples. We make the full dataset publicly available [1] and include additional labels to benefit further research, such as the pieces’ bounding boxes.

## 4. Proposed Method

This section details the pipeline’s three main stages: (i) board localisation, (ii) occupancy classification, (iii) piece classification, and then presents a transfer learning approach for adapting the system to unseen chess sets. The main idea is as follows: we locate the pixel coordinates of the four chessboard corners and warp the image so that the chessboard forms a regular square grid to eliminate perspective distortion in the sizes of the chess squares before cropping them. Then, we train a binary CNN classifier to determine individual squares’ occupancies and finally input the occupied squares (cropped using taller bounding boxes) to another CNN that is responsible for determining the piece types. To adapt to a previously unseen chess set, the board localisation algorithm can be reused without modification, but the CNNs must be fine-tuned on two images of the new chess set.

### 4.1. Board Localisation

To determine the location of the chessboard’s corners, we rely on its regular geometric nature. Each square on the physical chessboard has the same width and height, even though their observed dimensions in the input image vary due to 3D perspective distortion. A chessboard consists of 64 squares arranged in an 8×8 grid, so there are nine horizontal and nine vertical lines.

#### 4.1.1. Finding the Intersection Points

The first step of the algorithm detects the majority of horizontal and vertical lines and finds their intersection points. We convert the image to greyscale and apply the Canny edge detector [25], the result of which is shown in Figure 2b. Next, we perform the Hough transform [26] in order to detect lines that are formed by the edges which typically yields around 200 lines, most of which are very similar. Therefore, we split them into horizontal and vertical lines and then eliminate similar ones. Experiments show that simply setting thresholds for the lines’ directions is insufficient for robustly classifying them as horizontal or vertical because the camera may be tilted quite severely. Instead, we employ an agglomerative clustering algorithm (a bottom-up hierarchical algorithm where each line starts off in its own cluster and pairs of clusters are continually merged in a manner that minimises the variance within the clusters), using the smallest angle between two given lines as the distance metric. Finally, the mean angle of both top-level clusters determines which cluster represents the vertical and which the horizontal lines (see Figure 2c).

To eliminate similar horizontal lines, we first determine the mean vertical line in the vertical cluster. Then, we find the intersection points of all the horizontal lines with the mean vertical line and perform a DBSCAN clustering [27] to group similar lines based on these intersection points, retaining only the mean horizontal line from each group as the final set of discovered horizontal lines. We apply the same procedure vice-versa for the vertical lines, and compute all intersection points.

#### 4.1.2. Computing the Homography

It is often the case that we detect fewer than nine horizontal and vertical lines (like in Figure 2d), thus we must determine whether additional lines are more likely to be above or below the known horizontal lines (and likewise to the left or right of the known vertical lines). Instead of computing where the candidate lines would be in the original image, it is easier to warp the input image so that the intersection points form a regular grid of squares (which must be done for cropping the samples for the occupancy and piece classifiers later anyway) and then to reason about that warped image because the missing lines will lie on that grid. This projective transformation is characterised by a homography matrix ***H*** that we find using a RANSAC-based algorithm that is robust even when lines are missing (or additional lines are detected from straight edges elsewhere in the image) and shall be described below:Randomly sample four intersection points that lie on two distinct horizontal and vertical lines (these points describe a rectangle on the chessboard).Compute the homography matrix ***H*** mapping these four points onto a rectangle of width $s_{x}=1$ and height $s_{y}=1$. Here, $s_{x}$ and $s_{y}$ are the horizontal and vertical scale factors, as illustrated in Figure 3.Project all other intersection points using ***H*** and count the number of inliers; these are points explained by the homography up to a small tolerance γ (i.e., the Euclidean distance from a given warped point (x,y) to the point (round(x),round(y)) is less than γ).If the size of the inlier set is greater than that of the previous iteration, retain this inlier set and homography matrix ***H*** instead.Repeat from step 1 for $s_{x}=2,3,\ldots ,8$ and $s_{y}=2,3,\ldots .,8$ to determine how many chess squares the selected rectangle encompasses.Repeat from step 1 until at least half of the intersection points are inliers.Recompute the least squared error solution to the homography matrix ***H*** using all identified inliers.

Next, we warp the input image and inlier intersection points according to the computed homography matrix ***H***, obtaining a result like in Figure 4a. The intersection points are quantised so that their *x* and *y* coordinates are whole numbers because each chess square is now of unit length. Let $x_{\min}$ and $x_{\max}$ denote the minimum and maximum of the warped coordinates’ *x*-components, and similarly $y_{\min}$ and $y_{\max}$ denote the same concept in the vertical direction.

If $x_{\max}-x_{\min}=8$, we detected all lines of the chessboard and no further processing is needed. When $x_{\max}-x_{\min}<8$, as is the case in Figure 4a, we compute the horizontal gradient intensities for each pixel in the warped image in order to determine whether an additional vertical line is more likely to occur one unit to the left or one unit to the right of the currently identified grid of points. To do so, we first convolve the greyscale input image with the horizontal Sobel filter in order to obtain an approximation for the gradient intensity in the horizontal direction (Figure 4b). Then, we apply Canny edge detection in order to eliminate noise and obtain clear edges (Figure 4c). Large horizontal gradient intensities give rise to vertical lines in the warped image, so we sum the pixel intensities in Figure 4c along the vertical lines at $x=x_{\min}-1$ and $x=x_{\max}+1$ (with a small tolerance to the left and to the right). Then, if the sum of pixel intensities was greater at $x=x_{\min}-1$ than at $x=x_{\max}+1$, we update $x_{\min}\leftarrow x_{\min}-1$, or otherwise $x_{\max}\leftarrow x_{\max}+1$. We repeat this processs until $x_{\max}-x_{\min}=8$. An analogous procedure is carried out for the horizontal lines with $y_{\min}$ and $y_{\max}$. Finally, these four values describe the two outer horizontal and vertical lines of the chessboard in the warped image. The optimal parameters for the Hough transform and Canny edge detectors described in this section are found using a grid search over sensible parameters on a small subset of the training set.

### 4.2. Occupancy Classification

We find that performing piece classification directly after detecting the four corner points with no intermediate step yields a large number of false positives, i.e., empty squares being classified as containing a chess piece (see Figure 5). To solve this problem, we first train a binary classifier on cropped squares to decide whether they are empty or not. Cropping the squares from the warped image is trivial because the squares are of equal size (see Figure 6).

We devise six vanilla CNN architectures for the occupancy classification task, of which two accept 100×100 pixel input images and the remaining four require the images to be of size 50×50 pixels. They differ in the number of convolutional layers, pooling layers, and fully connected layers. When referring to these models, we use a 4-tuple consisting of the input side length and the three aforementioned criteria. The final fully connected layer in each model contains two output units that represent the two classes (occupied and empty). Figure 7 depicts the architecture of cnn (100,3,3,3) which achieves the greatest validation accuracy of these six models. Training proceeds using the Adam optimiser [28] with a learning rate of 0.001 for three whole passes over the training set using a batch size of 128 and cross-entropy loss.

Apart from the vanilla architectures, we fine-tune deeper models (VGG [29], ResNet [30], and AlexNet [16]) that were pre-trained on the ImageNet [31] dataset. The final layer of each pre-trained model’s classification head is replaced with a fully-connected layer that has two output units to classify ‘empty’ and ‘occupied’ squares. Due to the abundance of data in the training set, we train the classification head for one epoch with a learning rate $\alpha =10^{-3}$ (while the other layers are frozen), followed by the whole network for two epochs with $\alpha =10^{-4}$.

### 4.3. Piece Classification

The piece classifier takes as input a cropped image of an occupied square and outputs the chess piece on that square. There are six types of chess pieces (pawn, knight, bishop, rook, queen, king), and each piece can either be white or black in colour, thus there are a dozen classes.

Some special attention is directed to how the pieces are cropped. Simply following the approach described in the previous section provides insufficient information to classify pieces. Consider for example the white king in Figure 5: cropping only the square it is located on would not include its crown which is an important feature needed to distinguish between kings and queens. Instead, we employ a simple heuristic that extends the height of bounding boxes for pieces further back on the board, and also the width depending on its horizontal location. As a further preprocessing step, we horizontally flip the bounding boxes of pieces on the left side of the board to ensure that the square in question is always in the bottom left of the image. This helps the classifier understand what piece is being referred to in samples where the larger bounding box includes adjacent pieces in the image. Figure 8 shows a random selection of samples generated in this way.

We train a total of six CNNs with 12 output units in the final layer for the piece classification task. For the pre-trained models, we follow the same two-stage training regime, but double the number of epochs at each stage compared to the previous section. Furthermore, we evaluate one more architecture, InceptionV3 [32], which shows a greater potential in light of this much more challenging task. The remaining two models are the two best-performing CNNs from the previous section. We pick the model with the highest accuracy score on the validation set to be used in the chess recognition pipeline.

### 4.4. Fine-Tuning to Unseen Chess Sets

Chess sets vary in appearance. CNNs trained on one chess set are likely to perform poorly on images from another chess set because the testing data is not drawn from the same distribution. Due to the inherent similarities in the data distributions (the fact that both are chess sets and that the source and target tasks are the same), we are able to employ a form of *few-shot* transfer learning; that is, using only a small amount of data in order to adapt the CNNs to the new distribution.

An advantage of our approach to board localisation is that it requires no fine-tuning to different chess sets because it employs conventional computer vision techniques such as edge and line detection. Therefore, we can fine-tune the CNNs *without* a labelled dataset: just two pictures of the starting position on the chessboard suffice (one from each player’s perspective) because the configuration of pieces is known (the starting position is always the same), as shown in Figure 9. We can localise the board and crop the squares using the method described in Section 4.1 to generate the training samples for fine-tuning the CNNs.

Since there are only two input images, the occupancy classifier must be fine-tuned using only 128 samples (each board has 64 squares). Moreover, the piece classifier has only 64 training samples because there are 32 pieces on the board for the starting position. While the CNN for occupancy detection is a binary classifier, the piece classifier must undertake the more challenging task of distinguishing between a dozen different piece types. Furthermore, the data is not balanced between the classes: for example, there are 16 training samples for black pawns, but only two for the black king, and some pieces are more difficult to detect than others (pawns usually look similar from all directions whereas knights do not). For the reasons listed above, we employ heavy data augmentations for the piece classifier at training time. We shear the piece images in such a way that the square in question remains in the bottom left of the input image. Of all augmentations, this transformation exhibits the most significant performance gains, likely due to its similarity to actual perspective distortion. We also employ random colour jittering (varying brightness, contrast, hue, and saturation), scaling, and translation. Figure 10 depicts a random sample of four outputs obtained by applying the augmentations to a cropped input image of the pawn on d2 in Figure 9a.

To fine-tune the networks, we follow a two-stage approach similar to the previous sections (where the models were pre-trained on ImageNet). First, we train only the classification head with a learning rate of 0.001, and then decrease the learning rate by a factor of ten and train all layers. In the case of the occupancy classifier, we perform 100 iterations over the entire training set at both stages, essentially following the same regime as Section 4.2. For the piece classifier, we execute an additional 50 iterations in the second stage to ensure reliable convergence. The loss curve is not as smooth as in the previous sections due to the small dataset, but training is still able to converge to a low loss value.

## 5. Results and Discussion

### 5.1. Board Localisation

To evaluate the performance of the board localisation algorithm, we counted a prediction as accurate if the distance of each of the four predicted corner points to the ground truth was less than 1% of the width of the input image. The algorithm produced inaccurate predictions in 13 cases out of 4400 samples in the training set, so its accuracy was 99.71%. The validation set of size 146 saw no mistakes and thus achieved an accuracy of 100.00%. As such, we concluded that the chessboard corner detection algorithm did not overfit as a result of the grid search.

### 5.2. Occupancy and Piece Classification

Each occupancy classifier was trained separately on the dataset of squares that were cropped to include contextual information (by increasing the bounding box by 50% in each direction, as explained in Figure 6), and again on the same samples except that the squares were cropped tightly. Key performance metrics for each model are summarised in Table 1. In each case, the model trained on the samples that contained contextual information outperformed its counterpart trained on tightly cropped samples, indicating that the information around the square itself was useful. The ResNet model achieved the highest validation accuracy (99.96%) of all evaluated architectures.

The fine-tuned deeper models performed better than the vanilla CNNs, although the differences in accuracy were small and every model achieved accuracies above 99%. This is likely due to the increased number of trainable parameters (up to two orders of magnitude higher than the simple CNNs), the use of transfer learning, and the more complex architectural designs. Nonetheless, it is evident in Table 1 by comparing the training and validation accuracies that none of the models suffered from overfitting which is not suprising given the size of the dataset. We selected the ResNet model for use in the chess recognition pipeline because it attained the highest validation accuracy score.

For piece classification, the results in Table 2 indicate a more significant difference between the hand-crafted CNNs and the deeper models (around three percentage points) than is the case for the occupancy classifier. The InceptionV3 model achieved the best performance with a validation accuracy of 100%, i.e., there were no misclassifications in the validation set, so we adopted that model in the chess recognition pipeline.

### 5.3. End-to-End Pipeline

The left side of Table 3 lists key evaluation metrics of the end-to-end pipeline on the train, validation, and test sets of the rendered dataset. There was no indication of overfitting because there were only slight differences in the results of the train and test sets. The two CNNs performed on par or even better on the held-out test set than the training set, and likewise did the corner detection algorithm. However, these differences—albeit slightly suprising—were negligible due to their insignificant magnitudes; in fact, the performance on the validation set was even higher than on the test set.

The end-to-end per-board accuracy on the test set was 93.86%, and when allowing just one mistake on the board, that accuracy increased to 99.71%. Comparing that first accuracy figure to the training set, there was a decrease of almost one percentage point which might seem peculiar because the three main stages each performed better or on par with the scores of the training set. However, this is explained by the fact that the system had more incidences with two or more misclassified squares in the training set than the test set.

The average number of misclassified squares per board lay at 0.15 on the test set as compared to 0.27 on the training set. The confusion matrix in Figure 11 facilitates a more detailed analysis of the mistakes. The last row and column, representing the class ‘empty square’, contain the greatest number of incorrect samples which is a result of the worse performance of the occupancy classifier compared to the piece classifier. However, one must also take into account that the occupancy classifier had a more difficult task in this regard, since it had to determine whether a square was empty even when it was occluded by a piece in front of it. The piece classifier (which had an accuracy of 99.99%) yielded only three errors: in two cases it confused a knight with a bishop, and in one case a pawn with a rook. These results on the test set clearly demonstrate that the chess recognition system was highly accurate.

For a chess recognition system to be practically effective, it must also be able to perform an inference in a reasonable amount of time. To test this, we recorded the execution time for each of the test set samples on a Linux machine with a quad-core 3.20 GHz Intel Core i5-6500 central processing unit (CPU) and a 6 GB NVIDIA GeForce GTX 1060 graphics processing unit (GPU). We conducted this experiment twice: once with GPU acceleration and once without. Figure 12 shows that the pipeline was around six times faster when utilising the GPU. This is to be expected because the forward pass through the neural network was optimised for parallel computation on a GPU. Therefore, execution time of the occupancy and piece classifiers was significantly lower on the GPU, whereas the board localisation (which ran on the CPU regardless) took a similar amount of time across both trials. Overall, the mean inference time was less than half a second on the GPU and just over 2 seconds on the CPU, although the latter measurement exhibited a significantly greater variance. The speed was sufficient for most practical purposes, and the GPU-accelerated pipeline may even be suited for real-time inference at two frames per second on just one GPU. Lastly, it should be noted that the occupancy classifier needed just a fraction of the time required by the piece classifier. This can be explained by the more complex architecture of the InceptionV3 [32] network as opposed to the ResNet [30] model and the greater input size which resulted in the number of parameters being twice as high in the piece classifier.

### 5.4. Unseen Chess Set

In order to evaluate the effectiveness of the approach outlined in Section 4.4, we created a dataset of images captured from a physical chess set. As explained earlier, the training set consisted only of the two pictures of the starting position that are depicted in Figure 9. The test dataset consisted of 27 images obtained by playing a game of chess (using the same chess set as in Figure 9) and taking a photo of the board after each move from the perspective of the current player. These samples were manually labelled with FEN strings describing the position.

While the fine-tuned occupancy classifier achieved a good validation accuracy out of the box, the piece classifier performed quite poorly without data augmentations at training time. The use of data augmentation resulted in a net increase in the accuracy of the position inference by 45 percentage points (from 44% without data augmentation to 89% with augmentation). Furthermore, the mean number errors per position decreased from 2.3 squares to 0.11 squares.

Key indicators for evaluating the performance of the chess recognition pipeline using the newly fine-tuned models on the transfer learning dataset are summarised in the two right columns of Table 3. The baseline approach (the chess recognition pipeline without fine-tuning to this new dataset) misclassified an average of 9.33 squares per board, whereas the fine-tuned system misclassified only 0.11 on the test set. Furthermore, the baseline identified no positions without mistakes whereas the fine-tuned system correctly identified 89% of the 27 positions. The remaining 11% of positions were identified with just one mistake. In other words, there were only three errors in all 27 samples in the test set. The results show how the use of transfer learning with careful data augmentation enabled the chess recognition pipeline to be adapted to a real-world chess set.

As a sanity check, we displayed a warning to the user if the predicted FEN string was illegal according to the rules of chess, prompting the user to retake the picture. All three of the incorrectly predicted positions were illegal positions, so these errors would be caught in practice. However, in the test set of the synthesised dataset, 21 of the 342 positions were identified incorrectly, and only one of these 21 predictions was illegal, so the other 20 errors would only be caught by manual inspection.

## 6. Summary and Conclusions

Motivated by the cumbersome process of transferring a chess position from the board to the computer in order to facilitate computer analysis, this paper presents an end-to-end chess recognition system that outperforms all existing approaches. It correctly predicts 93.86% of the chess positions without any mistakes, and when permitting a maximum of one mistake per board, its accuracy lies at 99.71%. The system achieves a per-square accuracy of 99.77% on the test set, thus reducing the per-square error rate of the current state of the art [15] by a factor of 28 from 6.55% to 0.23%. However, due to the lack of any published datasets for chess recognition (an issue recognised by several others [11,14,15]), we were unable to evaluate our system on their datasets, and could thus only compare the reported accuracy scores. To benefit future research and facilitate fair benchmarks, we put forth a new and much larger synthesised dataset with rich labels (containing over 3000 samples for each piece type as compared to 200 in Mehta et al.’s dataset [15]) and make it available to the public [1].

Furthermore, we develop the first few-shot transfer learning approach for chess recognition by demonstrating that with only two photos of a new chess set, the pipeline can be fine-tuned to a previously unseen chess set using carefully chosen data augmentations. On a per-square basis, that fine-tuned algorithm reaches an error rate of 0.17%, even surpassing the accuracy of the current state of the art system mentioned above which was trained on a lot more than two images. All code used to run experiments is available so that the results can be reproduced independently.

## Figures and Tables

**Figure 1 jimaging-07-00094-f001:**
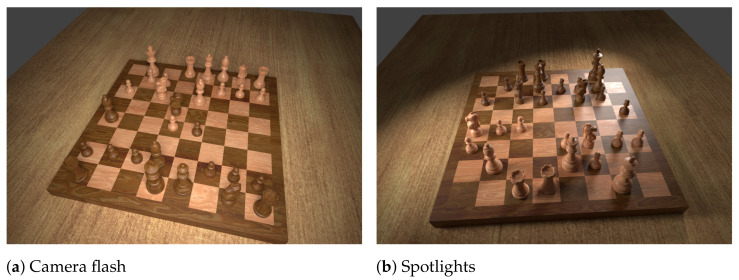
Two samples from the synthesised dataset showing both lighting modes.

**Figure 2 jimaging-07-00094-f002:**
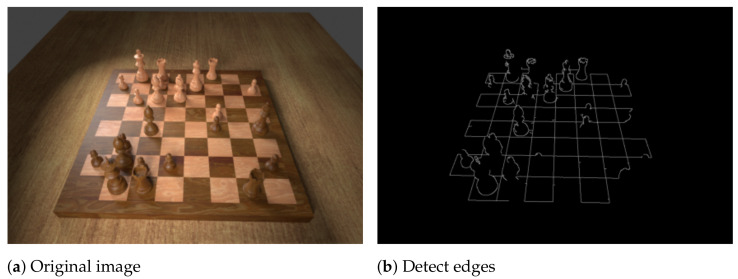
The process of determining the intersection points on the chessboard.

**Figure 3 jimaging-07-00094-f003:**
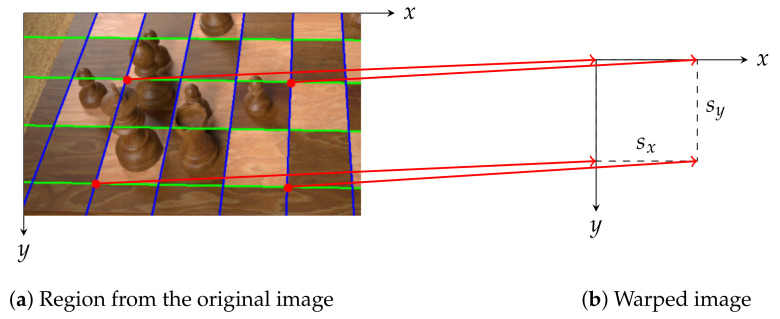
Four intersection points are projected onto the warped grid. The optimal values for the scale factors *s_x_* and *s_y_* are chosen based on how many other points would be explained by that choice, in order to determine the actual number of horizontal and vertical chess squares in the rectangular region from the original image. In this example, the algorithm finds *s_x_* = 3 and *s_y_* = 2.

**Figure 4 jimaging-07-00094-f004:**
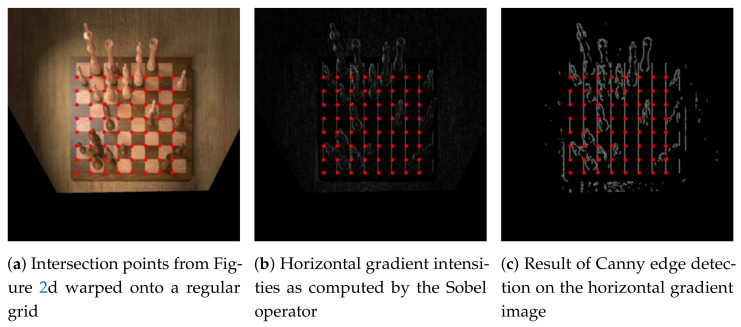
Horizontal gradient intensities calculated on the warped image in order to detect vertical lines. The red dots overlaid on each image correspond to the intersection points found previously. Here, *x_max_* − *x_min_* = 7 because there are eight columns of points instead of nine (similarly, the topmost horizontal line will be corrected by looking at the vertical gradient intensities).

**Figure 5 jimaging-07-00094-f005:**
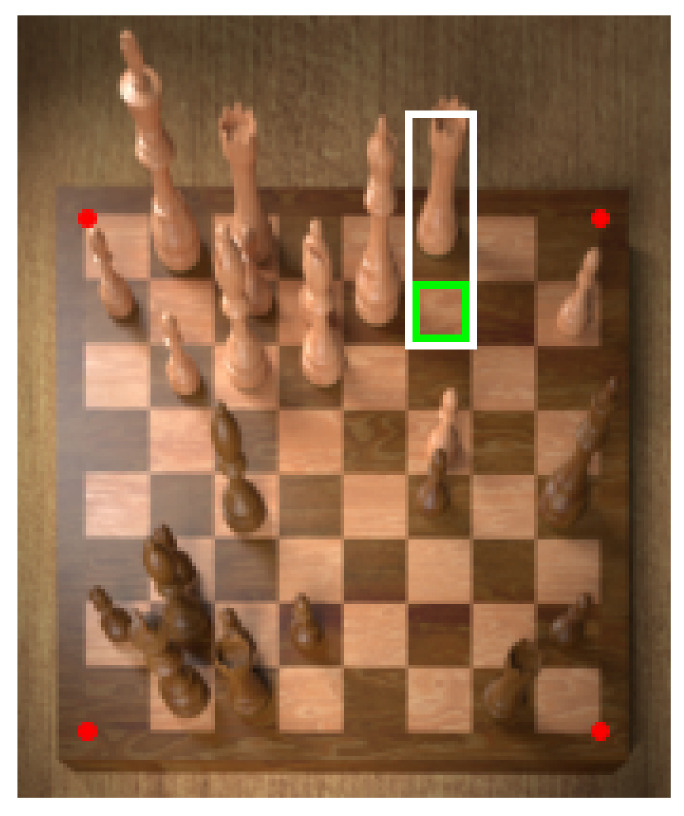
An example illustrating why an immediate piece classification approach is prone to reporting false positives. Consider the square marked in green. Its bounding box for piece classification (marked in white) must be quite tall to accomodate tall pieces like a queen or king (the box must be at least as tall as the queen in the adjacent square on the left). The resulting sample contains almost the entire rook of the square behind, leading to a false positive.

**Figure 6 jimaging-07-00094-f006:**
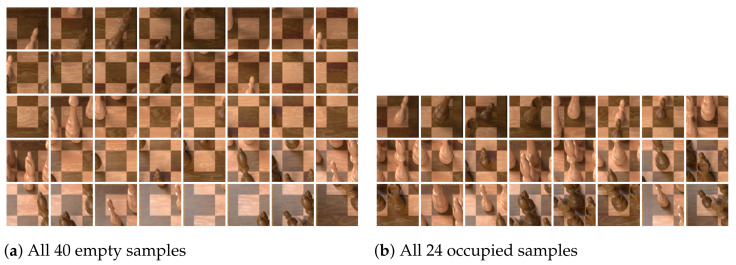
Samples for occupancy classification generated from the running example chessboard image. The squares are cropped with a 50% increase in width and height to include contextual information.

**Figure 7 jimaging-07-00094-f007:**
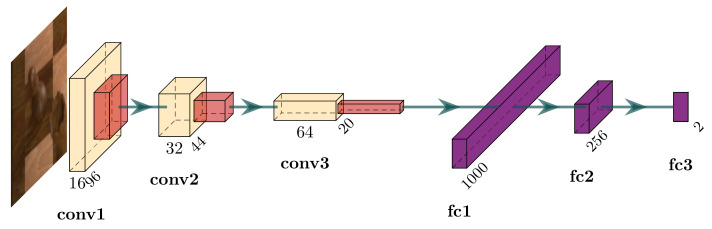
Architecture of the CNN (100,3,3,3) network for occupancy classification. The input is a three-channel RGB image with 100×100 pixels. The first two convolutional layers (yellow) have a kernel size of 5×5 and stride 1 and the final convolutional layer has a kernel size of 3×3. Starting with 16 filters in the first convolutional layer, the number of channels is doubled in each subsequent layer. Each convolutional layer uses the rectified linear unit (ReLU) activation function and is followed by a max pooling layer with a 2×2 kernel and stride of 2. Finally, the output of the last pooling layer is reshaped to a 640,000-dimensional vector that passes through two fully connected ReLU-activated layers before reaching the final fully connected layer with softmax activation.

**Figure 8 jimaging-07-00094-f008:**
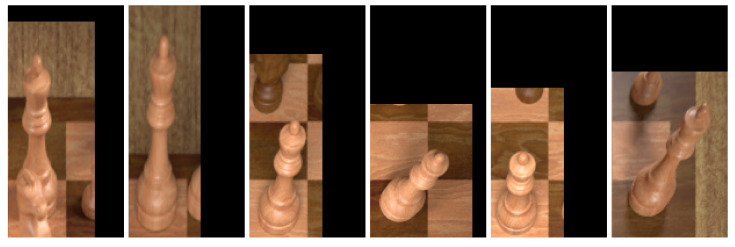
A random selection of six samples of white queens in the training set. Notice that the square each queen is located on is always in the bottom left of the image and of uniform dimensions across all samples.

**Figure 9 jimaging-07-00094-f009:**
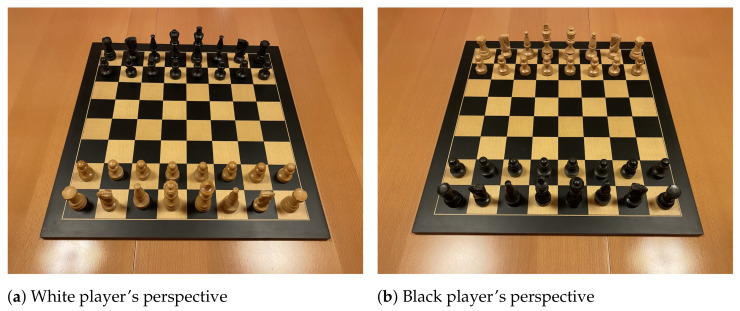
The two images of the unseen chess set used for fine-tuning the chess recognition system. The images require no labels because they show the starting position from each player’s perspective, thus the chess position is known. Note that unlike the large dataset used for initial training, this dataset contains photos of a real chessboard, as opposed to rendered images.

**Figure 10 jimaging-07-00094-f010:**
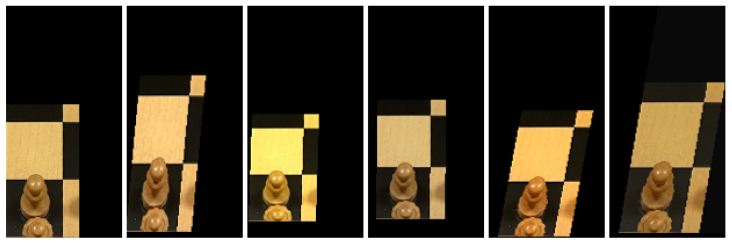
The augmentation pipeline applied to an input image (left). Each output looks different due to the random parameter selection.

**Figure 11 jimaging-07-00094-f011:**
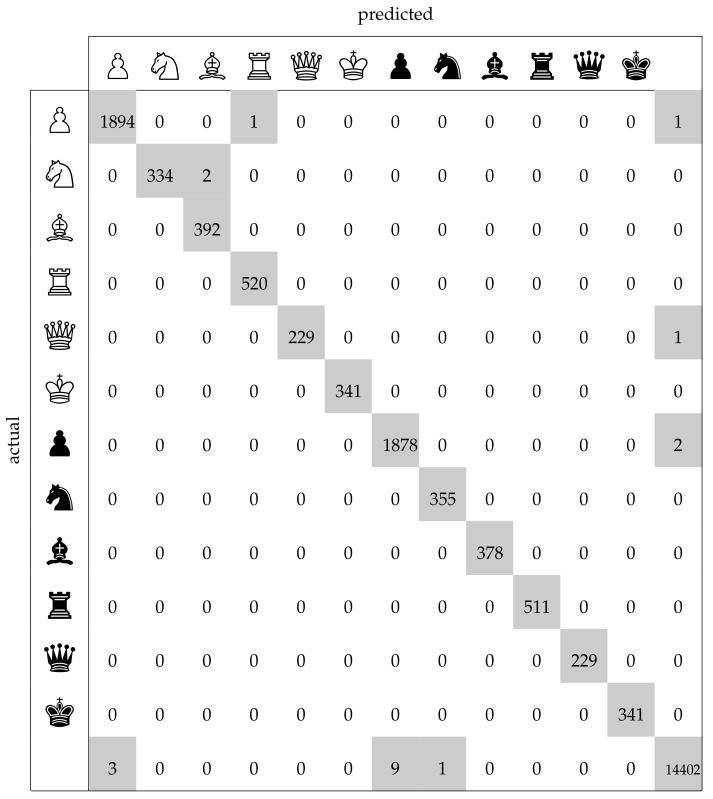
Confusion matrix of the per-square predictions on the test set. Non-zero entries are highlighted in grey. The final row/column represents empty squares. Chessboard samples whose corners were not detected correctly are ignored here.

**Figure 12 jimaging-07-00094-f012:**
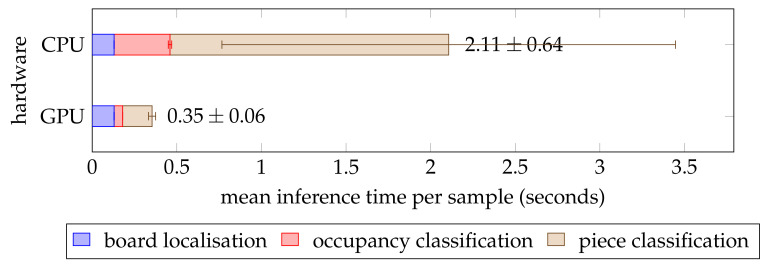
Inference time benchmarks of the chess recognition pipeline on the test set, averaged per sample. The error bars indicate the standard deviation. All benchmarks were carried out on the same machine, although the data for the trial labelled cpu was gathered without gpu acceleration.

**Table 1 jimaging-07-00094-t001:** Performance of the trained occupancy classifiers. Models prefixed with “cnn” are vanilla CNNs where the 4-tuple denotes the side length of the square input size in pixels, the number of convolution layers, the number of pooling layers, and the number of fully connected layers. The check mark in the left column indicates whether the input samples contained contextual information (cropped to include part of the adjacent squares). We report the total number of misclassifications on the validation set (consisting of 9346 samples) in the last column. The differences between training and validation accuracies indicate no overfitting.

	Model	# Trainable Parameters	Train Accuracy	Val Accuracy	Val Errors
**✓**	ResNet [30]	1.12 × 10^7^	99.93%	99.96%	4
**✓**	VGG [29]	1.29 × 10^8^	99.96%	99.95%	5
**✗**	VGG [29]	1.29 × 10^8^	99.93%	99.94%	6
**✗**	ResNet [30]	1.12 × 10^7^	99.94%	99.90%	9
**✓**	AlexNet [16]	5.7 × 10^7^	99.74%	99.80%	19
**✗**	AlexNet [16]	5.7 × 10^7^	99.76%	99.76%	22
**✓**	CNN (100, 3, 3, 3)	6.69 × 10^6^	99.70%	99.71%	27
**✓**	CNN (100, 3, 3, 2)	6.44 × 10^6^	99.70%	99.70%	28
**✗**	CNN (100, 3, 3, 2)	6.44 × 10^6^	99.61%	99.64%	34
**✓**	CNN (50, 2, 2, 3)	4.13 × 10^6^	99.62%	99.59%	38
**✓**	CNN (50, 3, 1, 2)	1.86 × 10^7^	99.67%	99.56%	41
**✓**	CNN (50, 3, 1, 3)	1.88 × 10^7^	99.66%	99.56%	41
**✓**	CNN (50, 2, 2, 2)	3.88 × 10^6^	99.64%	99.54%	43
**✗**	CNN (50, 2, 2, 3)	4.13 × 10^6^	99.57%	99.52%	45
**✗**	CNN (100, 3, 3, 3)	6.69 × 10^6^	99.55%	99.50%	47
**✗**	CNN (50, 3, 1, 2)	1.86 × 10^7^	99.44%	99.50%	47
**✗**	CNN (50, 2, 2, 2)	3.88 × 10^6^	99.54%	99.44%	52
**✗**	CNN (50, 3, 1, 3)	1.88 × 10^7^	99.41%	99.39%	57

**Table 2 jimaging-07-00094-t002:** Performance of the trained piece classifiers.

Model	# Trainable Parameters	Train Accuracy	Val Accuracy	Val Errors
InceptionV3 [32]	2.44 × 10^7^	99.98%	100.00%	0
VGG [29]	1.29 × 10^8^	99.84%	99.94%	2
ResNet [30]	1.12 × 10^7^	99.93%	99.91%	3
AlexNet [16]	5.71 × 10^7^	99.51%	99.02%	31
CNN (100, 3, 3, 2)	1.41 × 10^7^	99.62%	96.94%	97
CNN (100, 3, 3, 3)	1.44 × 10^7^	99.49%	99.49%	98

**Table 3 jimaging-07-00094-t003:** Performance of the chess recognition pipeline on the train, validation, and test datasets, as well as the fine-tuned pipeline on the unseen chess set.

	Rendered Dataset			Unseen Chess Set	
Metric	Train	Val	Test	Train	Test
mean number of incorrect squares per board	0.27	0.03	0.15	0.00	0.11
percentage of boards predicted with no mistakes	94.77%	97.95%	93.86%	100.00%	88.89%
percentage of boards predicted with ≤1 mistake	99.14%	99.32%	99.71%	100.00%	100.00%
per-square error rate	0.42%	0.05%	0.23%	0.00%	0.17%
per-board corner detection accuracy	99.59%	100.00%	99.71%	100.00%	100.00%
per-square occupancy classification accuracy	99.81%	99.97%	99.92%	100.00%	99.88%
per-square piece classification accuracy	99.99%	99.99%	99.99%	100.00%	99.94%

## Data Availability

The synthesised dataset presented in this article is openly available in the Open Science Framework at 10.17605/OSF.IO/XF3KA [1].

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
