# Peer review of "Determining Chess Game State from an Image"

_2313-433X, 2021, doi:10.3390/jimaging7060094_

Round 1

Reviewer 1 Report

Overall this paper is well-written and presents a modern approach to the challenge of reading chess boards and positions. The overall approach is reasonably sound and leverages modern machine learning tools to do most of the heavy lifting of classification. Using a synthetic dataset is a good idea for creating a large dataset quickly, although I wondered why the authors didn't create an even larger dataset given that it should be easy to programmatically generate positions, angles, and lighting conditions.

My primary concern with the paper is around the experiment on an unseen chess set. Testing with only one other chess set for one 27-move match is insufficient to make a strong claim that the system works generally. I would recommend that the authors weaken the claim that they are making - the described experiment is still good data, but not "very strong and a testament to the effectiveness of using transfer learning.

I would also suggest that the authors consider a rules engine to check the validity of a FEN as well, which could improve the accuracy of the system further. There are some board positions that are illegal given the rules of chess, and some basic checks before providing an output could provide a prompt to the user to take another photo, rather than outputting an erroneous FEN.

I would also suggest that the authors may be in fact giving too much detail on the methodology in the paper (particularly for board localisation), but leave a decision on length to the editors.

Reviewer 2 Report

The authors proposed a method for determining chess game state from an image, which includes: board localisation, occupancy classification, and piece classification. The authors had also prepared a new dataset for chess board and pieces recognition consisting of rendered chessboard images, based on which the recognition method was validated. The dataset is CGI-based, however, it emulates real chessboard photos pretty well because of good quality of 3D models of pieces and board, different positions of pieces, camera angles, and lighting setups. The dataset contains 4,888 configurations of chess pieces, while the actual number of valid configurations is much greater. However, these positions are properly chosen based on games played by current World Chess Champion.

Chess game state recognition is maybe not among the most important computer vision subjects, but it may be useful for some chess players. Generally, the paper is very well written. The computer vision and machine learning methods seem valid (based on their description). They are also properly validated using the developed CGI dataset and unseen real photos of chess with transfer learning.

In my opinion the paper is suitable for submission. Only two minor comments should be addressed by the authors:

  1. Please explain in the paper what means the size Sx = 1 and Sy = 1 (line 187). It's not a metric unit or the number of pixels. I think I can guess what the values (lengths) 1, 2, 3, …, 8 mean, but it may not be clear to readers.
  2. It would be good to test computational time of the developed method and report results.
